# LncMyoD Promotes Skeletal Myogenesis and Regulates Skeletal Muscle Fiber-Type Composition by Sponging miR-370-3p

**DOI:** 10.3390/genes12040589

**Published:** 2021-04-17

**Authors:** Peiwen Zhang, Jingjing Du, Xinyu Guo, Shuang Wu, Jin He, Xinrong Li, Linyuan Shen, Lei Chen, Bohong Li, Jingjun Zhang, Yuhao Xie, Lili Niu, Dongmei Jiang, Xuewei Li, Shunhua Zhang, Li Zhu

**Affiliations:** 1College of Animal Science and Technology, Sichuan Agricultural University, Chengdu 611130, China; zpw1995@stu.sicau.edu.cn (P.Z.); b20171908@stu.sicau.edu.cn (J.D.); guoxinyu@stu.sicau.edu.cn (X.G.); wushuang@stu.sicau.edu.cn (S.W.); hejin19960812@163.com (J.H.); lixinrong0807@163.com (X.L.); shenlinyuan0815@163.com (L.S.); chenlei815918@sicau.edu.cn (L.C.); libohong@stu.sicau.edu.cn (B.L.); zhangjingjun@stu.sicau.edu.cn (J.Z.); 201604990@stu.sicau.edu.cn (Y.X.); dkynli@sicau.edu.cn (L.N.); jiangdm@sicau.edu.cn (D.J.); xuewei.li@sicau.edu.cn (X.L.); zhangshunhua@sicau.edu.cn (S.Z.); 2Farm Animal Genetic Resources Exploration and Innovation Key Laboratory of Sichuan Province, Sichuan Agricultural University, Chengdu 611130, China

**Keywords:** ceRNA, lncMyoD, mir-370-3p, fiber-type, myogenesis

## Abstract

The development of skeletal muscle is a highly ordered and complex biological process. Increasing evidence has shown that noncoding RNAs, especially long-noncoding RNAs (lncRNAs) and microRNAs, play a vital role in the development of myogenic processes. In this study, we observed that lncMyoD regulates myogenesis and changes myofiber-type composition. miR-370-3p, which is directly targeted by lncMyoD, promoted myoblast proliferation and inhibited myoblast differentiation in the C2C12 cell line, which serves as a valuable model for studying muscle development. In addition, the inhibition of miR-370-3p promoted fast-twitch fiber transition. Further analysis indicated that acyl-Coenzyme A dehydrogenase, short/branched chain (ACADSB) is a target gene of miR-370-3p, which is also involved in myoblast differentiation and fiber-type transition. Furthermore, our data suggested that miR-370-3p was sponged by lncMyoD. In contrast with miR-370-3p, lncMyoD promoted fast-twitch fiber transition. Taken together, our results suggest that miR-370-3p regulates myoblast differentiation and muscle fiber transition and is sponged by lncMyoD.

## 1. Introduction

Mammalian skeletal muscle is a type of heterogeneous organ [1]. An increasing number of studies have shown that skeletal muscle serves multiple functions in addition to movement, including secreting regulatory factors for crosstalk with distant organs and executing metabolic organ function [2,3]. Thus, skeletal muscle dysfunction may cause human diseases and ultimately damage individual health [4,5,6]. Muscle progenitor cells undergo a highly coordinated process during myogenesis, including myoblast proliferation, differentiation, and cell fusion into multinucleated myotubes. In addition, mammalian muscle fibers can generally divide into fast-twitch fibers (types IIb and IIx) and slow-twitch fibers (types I and IIa) [7]. The heterogeneity of muscle fibers is the basis of muscle function. Furthermore, growing evidence indicates that fiber-type composition is also closely associated with livestock meat quality [7,8,9]. Therefore, uncovering the mechanism of myoblast differentiation and changes in fiber-type composition is necessary for both humans and livestock.

Recently, with the development of high-throughput sequencing technologies, an increasing number of noncoding RNAs have been identified. Long noncoding RNAs (lncRNAs) are a class of noncoding RNAs that are marked by lengths longer than 200 bp [10]. Currently, growing evidence has shown that lncRNAs participate in gene expression regulation at the epigenetic level in a variety of vital processes, such as cancer, lipid metabolism, diabetes, and muscle development [11,12,13]. For example, the lncRNA Dum (developmental pluripotency-associated 2 (Dppa2) upstream binds muscle lncRNA) is a Dppa2 upstream transcript that silences Dppa2 by recruiting Dnmts (Dnmt3a, Dnmt3b, and Dnmt1) to the Dppa2 promoter region and then promotes myoblast differentiation and muscle regeneration [14]. Wakkach et al. [15] reported that the lncRNA Maxl AS could interact with Max1 pre-mRNA and attenuate Max1 function, which represses myoblast differentiation. Several researches reported that lncRNA play an important role in the formation of muscle, such as lncH19 [16], lncDum [17], lncMyoD, and lincYY1 [18]. In addition, regulatory factors target mesodermal cells and induces expression of MyoD and Myf5, which triggers extraocular muscle formation. LncMyoD which is a long noncoding RNA located away from MyoD locus, during myogenesis lncMyoD function as a competing endogenous RNA (ceRNA) binding IGF2-mRNA-binding protein 2 (IMP2) and regulating myoblast differentiation. Recently, Dong et al. [19] found that in muscle stem cells lncMyoD only binds with MyoD and does not bind with other myogenic regulator factors. Loss of lncMyoD prevents the establishment of a permissive chromatin environment at a myogenic E-box containing region, which is a key mechanism in the control of gene expression. However, it has not been fully elucidated in terms of the function and mechanism in myogenesis and fiber-type composition of lncMyoD. Therefore, it is of great significance to study the role of lncMyoD in regulating the myogenesis mechanism and fiber-type composition.

Increasing evidence demonstrates that lncRNAs also act as ceRNAs by sponging miRNAs to activate target transcripts. For example, the lncRNA Malat1 (metastasis-associated lung adenocarcinoma transcript 1) competes with miR-133 as a ceRNA to promote serum response factor expression and muscle cell differentiation [20]. miR-370-3p expressed a low expression level in cancer tissue, which was reported as a tumor suppressor [21,22]. miR-370-3p has been indicated to participate in cervical cancer, thyroid and bladder cancer, and so on. However, the molecular mechanisms of myogenesis still needs to be verified.

In this study, we focused on the relationship between lncMyoD and miR-370-3p in muscle development and revealed the mechanism by which lncMyoD acts as a ceRNA. In our study, we found that lncMyoD negatively regulated miR-370-3p and finally contributed to acyl-Coenzyme A dehydrogenase, short/branched chain (ACADSB) expression, thereby changing the muscle fiber-type composition.

## 2. Materials and Methods

### 2.1. Ethics Statement

The study was conducted under the approval of the Ethics Committee of Sichuan Agriculture University. All animal care and procedures performed in this study were conducted according to the Guidelines for Animal Experiments of Sichuan Agriculture University (Approval No. 20200037).

### 2.2. Cell Culture and Transfection

Transfection was performed using the method described previously [23]. Briefly, mouse C2C12 cells were maintained in Dulbecco’s modified Eagle’s medium (DMEM, Gibco, Thermo Fisher Scientific, Waltham, MA, USA) with 10% fetal bovine serum (FBS, Gibco, original U.S.) and incubated at 37 °C in 5% carbon dioxide. When the cell density reached approximately 80%, the cells needed to be subcultured. C2C12 myoblasts were seeded in 12-well plates, and the medium was switched to differentiation medium when the cell density reached 30% (proliferation) or 70%–80% (differentiation). Lipofectamine 3000 (Invitrogen, Carlsbad, CA, USA) was used for transfection. In addition, the differentiation medium contained 2% horse serum (Gibco, original U.S.). miR-370-3p mimics, inhibitor, mimic control, inhibitor control, si-ACADSB, si-lncMyoD, and si-negative control were transfected into C2C12 myoblasts (40 pmol/mL). The medium was changed after 6 h. The proliferation and differentiation media were changed every 24 h.

### 2.3. Animals and Collection of Muscles

In this study, two groups of C57B/L6 mice were fed a normal chow diet. Each experiment had a control group, and mice were randomly grouped. Muscle injury and regeneration were induced by 10 μM CTX. In brief, 8-week-old mice were injected with CTX, and the negative control group mice were injected with PBS. The anterior tibial muscle was collected for qRT-PCR analysis after 1, 3, 5, 7, and 14 days of injection (*n* = 6 in each group). Mice with increased expression of lncMyoD were generated as described by Kawase et al. [24] (*n* = 12 in each group). In brief, mice were injected with naked plasmids (20 ng per mice), and the plasmid was diluted with PBS. The injections were repeated every 3 days. In order to establish a muscle atrophy model, mice were subjected to dexamethasone (1 mg/kg) for 2 weeks, three times a week (*n* = 10 in each group). The Mdx mice were presented by Max Planck Institutes (*n* = 6 in each group).

### 2.4. Cell Proliferation Assay

Cell proliferation was assessed by Cell Counting Kit 8 (CCK-8, Zhuangmeng, Beijing, China) assay and 5-ethynyl-20-deoxyuridine (EdU) proliferation assay. Briefly, for the CCK-8 assay, C2C12 cells were seeded in 96-well plates. After transfection with mimics, inhibitor, negative control, siRNA, and siRNA negative control, we used CCK-8 (Zhuangmeng, Beijing, China) to measure cell proliferation through absorbance detection at 450 nm. The detection timepoints were 0, 24, 48, and 96 h after transfection.

For the 5-ethynyl-20-deoxyuridine (EdU) assay, C2C12 cells were seeded in 96-well plates. The cells were transfected, and the medium was collected after 6 h. Next, the cells were supplied with growth medium for 36 h. Then, 100 mL of 50 mM Edu (Ribio, Guangzhou, China) reagent was added to every well for 2 h. After staining, the images were captured using a Nikon TE2000 microscope (Nikon, Tokyo, Japan).

### 2.5. Immunofluorescence Analysis

As previously described [25], well-differentiated C2C12 cells were washed with PBS three times and fixed in 4% paraformaldehyde for 2 h. After washing with PBS three times, the cells were permeabilized with 0.5% Triton X-100 for 5 min and blocked with 2% goat serum (Beyotime, Guangzhou, China). Then, the cells were incubated for 24 h with an anti-MYHC antibody, anti-fast myosin antibody, and anti-slow myosin antibody. ImageJ software was used to measure the fluorescence intensity. The images were captured using a Nikon TE2000 microscope (Nikon, Tokyo, Japan).

### 2.6. Isolation of RNA and RT-PCR

Total RNA from tissues and cell samples was isolated with RNAiso Plus (TaKaRa, Dalian, China). cDNA was made using a PrimeScript TM RT Reagent Kit with gDNA Eraser (TaKaRa, Dalian, China) and Mir-X miRNA First-Strand Synthesis Kit (TaKaRa, Dalian, China). Quantitative real-time PCR (qRT-PCR) reactions were performed by using a SYBR Premix Ex Taq Kit (TaKaRa, Dalian, China) and a CFX96 real-time PCR detection system (Bio-Rad, Hercules, CA, USA). The primer sequences used for qRT-PCR are listed in Appendix A.

### 2.7. Luciferase Reporter Assay

The luciferase reporter plasmids psi-CHECK2TM (Promega, Madison, WI, USA) containing wild-type 3′UTR or mutant-type 3′UTR of ACADSB and lncMyoD were constructed by TsingKe Biotech (Chengdu, Sichuan, China). We then amplified the plasmid and purified it for later use. At the same time, we prepared the corresponding empty plasmid control for purification and use. We cultivated Hela cells and inoculated them in 96-well plates, where they were allowed to grow for 10–24 h (80% confluence). The reporter gene plasmid and miR-370-3p mimics or negative control co-transfected cells. We added 10 ng plasmid per well, 0.2 microliters of mirna transfection reagent (concentration 20 nM) per well, which was first incubated with opti-MEM alone for 5 min; then, we mixed the transfection reagent and plasmid with lipo3000 and incubated them for 15 min together, adding a corresponding hole for each orifice plate. We replaced this with fresh culture medium at 6 h.

After 36 h, HeLa cells were harvested, and luciferase activities were detected by a Dual-Glo Luciferase Assay System (Promega, Madison, WI, USA). We attached the entire DNA sequences of the vectors, as shown in Appendix A.

### 2.8. RIP Assay

The RIP assay was performed according to the EZ-Magna RIP Kit (Millipore). The cells (~2.0 × 10^7^) were lysed with RIP lysis buffer after transfection of miR-370-3p mimics and miR-370-3p mimics negative control with Lipofectamine 3000 for 48 h. Then, the mixture of RIP lysis buffer, magnetic beads with anti-AGO2 or negative control IgG, and RIP immunoprecipitation buffer was incubated with rotation overnight at 4 °C, and the expression of lncMyoD was detected by qRT-PCR after the RNA was purified.

### 2.9. Western Blotting

Lysis buffer was used to extract C2C12 protein, which was electrophoresed on SDS-PAGE and transferred to a 0.45 μm PVDF membrane (Bio-Rad). After transfer, each membrane was blocked with 5% nonfat milk in TBST buffer for 1–2 h and then incubated with the primary antibodies at 4 °C overnight, followed by a secondary antibody. The target bands were detected by ECL (BioSharp, Hefei, China).

### 2.10. Statistical Analysis

The relative expression levels of mRNA and miRNA were calculated using the 2−∆∆Ct method. Additionally, U6 and β-Actin and U6 snRNA were used as the internal reference genes. Each experiment was repeated at least three times with three biological repeats. Data were analyzed with Prism 8.0 version. All data are presented as the means ± standard deviation (S.D.). Differences in groups were analyzed with Student’s *t*-test when there were fewer than three experimental groups. One-way ANOVA was used when there were more than three experimental groups. *p* < 0.05 was considered to be statistically significant, * *p* < 0.05, ** *p* < 0.01.

## 3. Results 

### 3.1. LncMyoD Is Associated with Skeletal Myogenesis

Increasing evidence suggests that long noncoding RNAs are increasingly implicated in various developmental and pathological processes. It has also been reported that lncMyoD is a direct target of MyoD and plays an important role during myoblast differentiation. We further investigated the function of lncMyoD during myogenesis. Thus, we analyzed the expression of lncMyoD in skeletal muscle from 1-day-old and 1- and 6-week-old mice, including the newborn, juvenile, sexually mature and mature stages. We found that lncMyoD sharply declined at 1 day to 3 weeks at the early stage of skeletal muscle development; however, it was significantly upregulated at 3–4 weeks when skeletal muscle was differentiated into mature muscle tissue (Figure 1A). This result suggests that lncMyoD may participate in skeletal muscle development, and we observed a dynamic process in skeletal muscle development. Then, to further prove that lncMyoD participated in myogenesis in vivo, we used three mouse models: mdx, Dex, and CTX. The pathological changes in muscle tissue were degeneration, and necrosis of muscle fibers was observed in the muscle tissue of mdx mice (Figure 1B). As shown in Figure 1B, the mdx mice had widened skeletal muscle space, different muscle fiber sizes, and aggregated nuclei. qRT-PCR analysis showed that the expression of lncMyoD was significantly downregulated in the mdx group compared to the WT group (*p* < 0.01). 

For further confirmation, we examined LncMyoD expression in tibialis anterior muscle (TA muscle) after Dex was injected into the abdominal cavity of mice. As shown in Figure 1C, after the injection of Dex, the expression of LncMyoD was also significantly decreased (*p* < 0.01). 

To further investigate the role of LncMyoD during myogenesis, we examined the expression of LncMyoD in the TA muscle after injection of CTX, which induced muscle injury, and simulated the myogenesis process in muscle (Figure 1D). The tendency of LncMyoD expression was similar to that identified in a previous study [26]. After injury, the expression increased first at 1 day when the muscle responded to injury, and the expression fluctuated slightly during the muscle repair process and finally returned to the normal level when the muscle was repaired completely after 7 days (Figure 1D). The above results demonstrate that LncMyoD indeed participates in myogenesis.

We speculate that lncMyoD may also participate in myogenesis in vivo. Simultaneously, for further verification, the expression of lncMyoD was increased as described in a previous in vivo study [24]. After the expression of lncMyoD was increased in mice (Appendix A), we found that the muscle fiber cross-sectional area in the overexpression group was significantly increased (Figure 1E). qPCR results showed that increased expression of lncMyoD in vitro also promoted the expression of genes involved in myogenic differentiation (Figure 1F). 

### 3.2. LncMyoD Regulates Myoblast Proliferation and Differentiation

To explore the role of lncMyoD in myoblast proliferation and differentiation, we used C2C12 cells as the cell model. lncMyoD expression during C2C12 myoblast proliferation and differentiation was examined by qRT-PCR. The results showed that the expression of lncMyoD was slightly decreased during the early stage of proliferation from 36 h to 12 h; however, its expression was strongly upregulated in the C2C12 differentiation stage (Figure 2A). Many studies have demonstrated that ki67 is an indicator of proliferative activity. In terminal differentiation, myoblast cells exit the proliferative cycle and initiate muscle-specific gene expression. To further explore the function of lncMyoD in myoblast proliferation, we performed immunofluorescence for ki67 using TA muscle cross-sections from the lncMyoD-increased group and negative control group mice. Increased expression of lncMyoD in mice markedly decreased the expression of ki67, as shown in Figure 2B,C. qRT-PCR analysis showed that CDK4, CDK6, and cyclin D1 were strongly downregulated in the group in which lncMyoD was increased in mice. To explore the potential connection between lncMyoD and myogenesis, we detected whether silencing lncMyoD and overexpressing lncMyoD affected myoblast proliferation. C2C12 myoblast cells were transfected with small interfering RNA (siRNA) and the pcDNA3.1-lncMyoD vector during myoblast cell proliferation. The results showed that after successful silencing of lncMyoD, the expression of lncMyoD was downregulated 4-fold, whereas after transfection of the pcDNA3.1-LncMyoD vector, the expression of lncMyoD was markedly upregulated (Figure 2D). The results of the EdU cell proliferation analysis show that silencing lncMyoD promoted an EdU-positive cell ratio, and overexpressing lncMyoD inhibited the EdU-positive ratio compared to its respective control group (Figure 2E,F). In addition, this result was consistent with the CCK-8 assay (Figure 2G). Both CCK-8 and EdU cell proliferation analyses were used to detect the effects of silencing and overexpressing lncMyoD. The results showed that, compared with the NC group, the inhibition of lncMyoD significantly promoted cell proliferation, and overexpression of lncMyoD showed the opposite trend. Consistent with the results shown in Figure 2G, qRT-PCR analysis demonstrated that the expression of cell cycle factors was upregulated in the siRNA lncMyoD group, and overexpression of lncMyoD significantly inhibited the expression levels of CDK4, CDK6, and cyclin D1 (Figure 2H). In previous studies, lncMyoD was reported to promote myogenesis. To verify these findings further, we transfected the pcDNA3.1-lncMyoD vector and si-lncMyoD into C2C12 myoblasts with differentiation medium. As shown in Figure 2I, the bright-field microscopy results demonstrated that lncMyoD overexpression significantly promoted C2C12 myoblast differentiation compared with the pcDNA3.1 control group, and si-lncMyoD showed the opposite trend. Myoblast differentiation and hypertrophy play important roles during myogenesis. It is known that myogenic regulatory factors (MRFs) are essential during myogenesis [28,29,30]. Thus, we detected the expression levels of MyoD, MyoG, Myf5, and MRF4 through qPCR assay after we transfected pcDNA3.1-lncMyoD vector and si-lncMyoD into C2C12 myoblasts. qRT-PCR results confirmed that overexpressed lncMyoD in C2C12 myotubes increased the MyoD, MyoG, Myf5, and MRF4 mRNA expression levels, and after knockdown of lncMyoD in C2C12 myotubes, the myotubes were significantly reduced in the si-lncMyoD group compared with the control group as well (Figure 2J). Compared with the control, lncMyoD inhibition reduced MYHC^+^ cells, and overexpression of lncMyoD showed the reverse trend (Figure 2K,L).

### 3.3. LncMyoD Regulates the Composition of Myofiber Type

Our findings above showed that lncMyoD suppressed myoblast proliferation and promoted differentiation. Therefore, it is reasonable to speculate that lncMyoD may also be associated with muscle fiber-type composition. To confirm this speculation, we first detected the expression of lncMyoD in the soleus (SOL) and tibialis anterior (TA) muscles, which are different muscle fiber types (Appendix A). The results showed that lncMyoD was highly expressed in TA, which is a typical fast myofiber type. In contrast, the relative expression in SOL muscle, which is a typical slow myofiber type, showed a striking decrease. In addition, we detected fast myosin and slow myosin in mouse skeletal muscle, which increased lncMyoD expression. Increased lncMyoD expression in mice significantly increased fast myosin fiber composition and downregulated slow myosin fiber composition, as shown in Figure 3A,B. Furthermore, we detected the expression of fast and slow myosin-related genes, and the results were consistent with the immunofluorescence results, which showed that increased lncMyoD expression upregulated some fast myosin fiber-related genes and downregulated some slow twitch fibers (Appendix A). To further confirm the results, C2C12 myoblasts were transfected with the pcDNA3.1-lncMyoD vector and si-lncMyoD. The same analysis was used to evaluate the effect of lncMyoD on fast and slow myosin composition in vivo. As expected, the immunofluorescence analysis showed the same trend, which promoted fast myosin-positive cells and downregulated the slow myosin-positive cell ratio (Figure 3C,D). Moreover, the result was also confirmed by qRT-PCR assay (Figure 1E). Muscle fiber-type composition is one of the key factors that affect muscle function. A previous study showed that, compared with fast-twitch fibers, slow-twitch fibers have a higher mitochondrial content. Thus, we further detected mitochondria in C2C12 myoblasts transfected with the pcDNA3.1-lncMyoD vector and si-lncMyoD. Consistent with these findings, the cells transfected with si-lncMyoD had abundant mitochondria (Figure 3E,F). Therefore, these results proved that lncMyoD participates in myogenesis and changes muscle fiber composition both in vitro and in vivo.

### 3.4. miR-370-3p Is Involved in Myogenesis and Regulates the Composition of Myofibers

The epigenetic mechanism of ceRNAs underlying the function of lncMyoD in regulating muscle fiber composition is not well characterized. Thus, we used the RegRNA and lncTar databases to predict target genes of lncMyoD. After selecting conserved target genes, we chose miR-370-3p, which is a highly conserved miRNA, as our target gene. In this study, we observed that the expression of miR-370-3p displayed an opposite trend when increasing or reducing lncMyoD (Figure 4A,B). During C2C12 myoblast proliferation and differentiation, the expression of lncMyoD and miR-370-3p showed an opposite trend (Figure 4C). We found that the expression of miR-370-3p was reduced when C2C12 myoblasts were transfected with the pcDNA3.1-lncMyoD vector and upregulated after transfection with si-lncMyoD. In addition, we constructed a wild-type and mutant reporter of lncMyoD and used a dual-luciferase reporter assay system to detect luciferase activity. As shown in Figure 4D, overexpression of miR-370-3p sharply downregulated luciferase activity. The results suggested that the two genes might have a direct regulatory relationship. Moreover, we performed an AGO2 immunoprecipitation (RIP) assay. As shown in Figure 4E, the binding between lncMyoD and miR-370-3p increased significantly following the transfection of miR-370-3p mimics. However, the role of miR-370-3p in muscle development and fiber-type composition has not been identified. Previous studies have shown that miR-370-3p participates in hepatic lipid metabolism and ischemia-reperfusion injury [31,32]. However, the epigenetic mechanism by which miR-370-3p regulates myogenesis has not been characterized. Thus, the potential role of miR-370-3p in skeletal muscle development should be explored. First, we transiently transfected C2C12 myoblasts with miR-370-3p mimics, inhibitor, and negative control separately. As shown in Appendix A, the transfection of miR-370-3p was successful, and the expression of miR-370-3p increased significantly. In contrast, the transfection of inhibitor downregulated endogenic miR-370-3p 4.5-fold. These data suggest that the expression level of miR-370-3p was successfully overexpressed or decreased. Subsequently, CCK-8 analysis was used to evaluate the effect of miR-370-3p on myoblast proliferation. As shown in Figure 4F, compared with the control group, the cell proliferation ability of C2C12 myoblasts was significantly promoted when miR-370-3p mimics were transfected, while the transfection of inhibitor had the opposite effect. We obtained the same results after analysis by EdU proliferation assay (Figure 4G,H). The results showed that, compared with the control group, the overexpression of miR-370-3p was accompanied by a marked increase in the EdU-positive cell ratio, whereas it decreased in the inhibitor group. Prior studies have noted the importance of cell cycle progression regulators, including cyclins and cyclin-dependent protein kinases (CDKs), which participate in G1/S and G2/M transformation in mammalian cells [33,34]. It would be important to determine whether miR-370-3p regulates cell cycle factors that influence C2C12 myoblast proliferation. qRT-PCR analysis showed that the expression levels of CDK4, CDK6, and cyclin D1 were sharply decreased in the transfected inhibition group, whereas gene expression was much lower in the mimic group (Appendix A). These results collectively suggest that miR-370-3p may promote C2C12 myoblast proliferation. Upon differentiation, mononucleated myoblasts withdraw from the cell cycle, and miR-370-3p was proven to promote C2C12 myoblast proliferation [7]. Interestingly, miR-370-3p expression gradually decreased after day 4 (Figure 4C). Additionally, such trends were observed by Du et al. [35], demonstrating that miR-143 inhibited C2C12 myoblast differentiation. Based on this, we speculate that miR-370-3p may inhibit the differentiation of myoblasts while increasing their proliferation. To elucidate the potential effect of miR-370-3p on C2C12 myoblast differentiation, we continuously transfected miR-370-3p mimics, inhibitor, and negative control into C2C12 myoblasts throughout the whole differentiation period. As shown in Appendix A, miR-370-3p mimics and inhibitor were transfected into C2C12 myoblasts. We measured the expression levels of key regulators of muscle development. qRT-PCR analysis showed that the inhibition of miR-370-3p significantly increased the expression of these genes compared with that in the NC group (Appendix A). Dong et al. [19] reported that lncMyoD may bind with MyoD and regulate MyoD accessibility to chromatin during myogenesis. To establish whether miR-370-3p would interfere with MyoD and limit the expression of MyoD, we detected MyoD protein expression level after miR-370-3p and si-MyoD treatment. The result showed that after miR-370-3p overexpression the MyoD protein expression level was not significantly decreased; however, there was a significant reduction in MyoD after transfecting siMyoD in C2C12 compared with si-NC (Appendix A). Furthermore, immunofluorescence detection of MYHC showed that the inhibition of miR-370-3p markedly promoted C2C12 myoblast differentiation and myotube formation compared with the negative control group, while the overexpression of miR-370-3p significantly reduced the formation of myotubes (Figure 4I,J). In addition, MYHC-positive cells in the miR-370 inhibition group were strongly upregulated compared with those in the negative control group. Taken together, these results suggest that miR-370-3p inhibits C2C12 myoblast differentiation. Here, we transfected C2C12 myoblasts in differentiation medium (DM) with both pcDNA3.1-lncMyoD and miR-370-3p mimics to further confirm that lncMyoD directly targets miR-370-3p to regulate myogenesis. The rescue assay indicated that transfection with mimics of miR-370-3p inhibited myoblast differentiation and was partially counteracted by lncMyoD overexpression. Furthermore, rescue assay was performed on transfection of C2C12 in DM. As shown in Appendix A, overexpressed lncMyoD could rescue the inhibitory effect of miR-370-3p during myoblast differentiation.

Based on the above results showing that miR-370-3p has an inhibitory effect on C2C12 myoblast differentiation and promotes proliferation targeted by lncMyoD, we next aimed to determine whether the expression of miR-370-3p affects skeletal muscle fiber-type composition. To clarify this speculation, we first detected whether miR-370-3p affects the genes related to fast- and slow-twitch fibers after transfection with miR-370-3p mimics, inhibitor, and negative control. As shown in Appendix A, the overexpression of miR-370-3p significantly promoted slow fiber-related gene expression, while the inhibition of miR-370-3p exhibited the opposite trend. To verify the obtained results, immunofluorescence analysis of fast and slow myosin was performed to further confirm the effect of miR-370-3p on muscle fiber composition. The results consistently showed that the number of fast myosin-positive cells sharply decreased after miR-370-3p inhibition, and the overexpression group presented the opposite trend compared with the NC group (Figure 4K,L). Therefore, these results collectively indicated that miR-370-3p participated in skeletal muscle fiber-type transition.

### 3.5. ACADSB Is a Direct Target of miR-370-3p

Noncoding miRNAs are implicated in various biological processes and generally regulate target gene expression by binding with the target mRNA 3′-untranslated region. To further explore the potential mechanism underlying miR-370-3p regulation of C2C12 myoblast proliferation and differentiation, we used TargetScan, miRwalk, PicTar, and miRDbase to predict the target genes of miR-370-3p. *ACADSB* plays a crucial role in fatty acid metabolism and is one of the candidate genes [35]. The identity of the binding site was investigated by homology analysis, which was conserved among species. In this study, we found that the expression level of *ACADSB* was downregulated after transfection of miR-370-3p into C2C12 myoblasts (Appendix A). Moreover, we built wild-type and mutant double fluorescent reporters and transfected them into HeLa cells to test whether miR-370-3p targeted *ACADSB* by binding to the 3′ UTR (Figure 5A). As expected, overexpression of miR-370-3p resulted in a sharp decrease in WT-*ACADSB* 3′ UTR reporter luciferase activity, and it had no effect on the MUT-*ACADSB* 3′ UTR reporter (Figure 5B). These data suggest that *ACADSB* is a target gene of miR-370-3p. Finally, we further used a Western blot assay to confirm that ACADSB was a target gene of miR-370-3p. The Western blot assay showed that, when C2C12 cells were transfected with miR-370-3p mimics, the protein level of ACADSB was significantly downregulated (Figure 5C).

To identify the potential role of ACADSB in myoblast proliferation and differentiation, we transfected C2C12 myoblasts with siRNA. As shown in Appendix A, the expression of ACADSB was markedly reduced by about 50%, and we successfully transfected si-ACADSB into C2C12 myoblasts. We used CCK-8 detection and EdU proliferation analysis to examine the effects of silenced ACADSB. The results showed that silencing ACADSB had no effect on cell proliferation (Appendix A). Consistently, the expression levels of CDK4, CDK6, and cyclinD1 were not significantly different after ACADSB silencing (Appendix A). Thus, the results showed that ACADSB had no effect on C2C12 proliferation. We further detected myogenic regulatory factors by qRT-PCR analysis, and the results suggested that si-ACADSB transfection significantly decreased the expression level compared with the control group (Appendix A). Additionally, we detected MYHC-positive cells and the fusion index, which was consistent with the qRT-PCR analysis results (Figure 5D,E). These results show that ACADSB has no effect on C2C12 myoblast proliferation and promotes differentiation. 

We then explored whether ACADSB plays a role in fiber-type transitions. In addition, immunofluorescence results showed that inhibition of ACADSB promoted a slow fiber positive ratio compared with the NC group (Figure 5F,G). qRT-PCR analysis results showed that ACADSB silencing inhibited fast skeletal fiber-related genes but reduced the expression of slow fiber type-related genes (Figure 5H). The above results suggest that miR-370-3p downregulates ACADSB to inhibit the composition of slow-twitch myofibers.

## 4. Discussion

The regulation of skeletal muscle development requires many regulatory factors. Skeletal muscle fiber-type composition is related to human exercise capacity, metabolism, blood circulation, and so on. Thus, clarifying the mechanisms of the regulation of skeletal muscle fibers is necessary. In the present study, we found that lncMyoD promotes ACADSB expression through repressive miR-370-3p during myoblast differentiation and myofiber composition. Muscle atrophy and weakness are common symptoms of most muscle diseases that suppress muscle development and influence muscle function. 

Skeletal muscle mainly originates from the somite evolved from the paraxial mesoderm. The primitive myogenic progenitor cells in the somite undergo two stages of primary myogenesis and secondary myogenesis, and finally develop into fetal muscle, and myogenic progenitor cells gradually evolve. They are skeletal muscle stem cells. After birth, the mass and volume of the newborn fetus’ muscles continue to grow until they develop into adult skeletal muscles. At the same time, the number of skeletal muscle stem cells gradually decreases at birth, and finally locates between the muscle fiber membrane and the basement membrane, forming a pool of adult skeletal muscle stem cells, that is, satellite cells. Adult skeletal muscle is the most abundant tissue in the human body. It shows strong plasticity and adaptability to the stress response caused by various injury stimuli. The adaptive response of adult skeletal muscle is mainly reflected in two aspects: the first is the damage of skeletal muscle itself, such as acute trauma and muscular dystrophy. In this case, satellite cells are activated to initiate the muscle formation process and regenerate new muscle fibers or repair of existing muscle fibers. The second aspect is the muscle morphology changes caused by factors other than the skeletal muscle itself, such as cachexia, long-term use of glucocorticoids, and denervation. In this case, the skeletal muscle exhibits a decrease in mass and volume small, that is, skeletal muscle atrophy. In this study, we tested the expression of lncMyoD in the two modes of skeletal muscle damage and external factors affecting skeletal muscle damage. The mdx mouse is a classic model for studying muscular dystrophy. The pathological features of mdx muscles are muscle fiber diameters ranging in size, muscle fiber hypertrophy, atrophy, degeneration, and obvious proliferation of connective tissue in the perimysium and endomysium. We investigated the expression of lncMyoD by which CTX caused pulmonary impairment. Both mdx mouse model and CTX mouse model belong to damage of skeletal muscle itself. Additionally, ongoing injection of dexamethasone (Dex), a glucocorticoid, can also cause muscle atrophy, which is proposed as a possible model of muscle atrophy. We found that when the muscles did not have the ability to repair, the expression of lncMyoD decreased significantly in the mdx group. When healthy mice received acute toxin damage, the expression of lncMyoD first increased and then decreased during the 14-day repair process. In the dexamethasone-induced external muscle injury model, the expression of lncMyoD also decreased significantly. These results indicate that lncMyoD is involved in the biological process of myogenesis.

The biological process of myogenesis is a highly ordered progression. In particular, myoblasts fuse to form syncytial myotubes and multinucleated myotubes late in development into fully differentiated muscle fibers, and muscle fibers have different types that vary in their metabolic preferences and eventually build a highly ordered skeletal muscle structure [25]. In addition, different muscle fiber types vary in their metabolic preference. Previous studies have reported that mammalian skeletal muscle fibers consist of two major groups, slow twitch and fast twitch, and slow muscles (type I fibers) rely mainly on oxidative phosphorylation, whereas fast muscles (type II fibers) are primarily glycolytic [7]. It has been determined that skeletal muscle fiber composition is associated with noncoding RNA regulation. Du et al. indicated that miR-351-5p inhibits LACTB and sponges lnc-mg to regulate myofiber composition. Wen et al. [26] found that miR-22-3p promotes slow twitch fiber switching to fast twitch fibers. Muscle fibers, including several muscle fiber types and different fibers, have different metabolic and functional characteristics.

Previously, Gong et al. [20] demonstrated that lncMyoD competitively binds to IGF2-mRNA-binding protein 2 (IMP2) and represses N-Ras and c-Myc, which are IMP2-related translations. The MyoD-lncMyoD-IMP2 pathway reveals the mechanism by which MyoD represses myoblast proliferation. Mesodermal cells differentiate into mononuclear myoblasts after cell proliferation and migration, and myoblasts differentiate and fuse into myotubes. Finally, myofibers are formed by the fusion of multinucleated myotubes. Our study revealed that lncMyoD inhibits myoblast proliferation, promotes myoblast differentiation, and regulates myofiber composition both in vitro and in vivo.

Numerous studies have shown that, during myoblast proliferation and differentiation, related miRNAs play a key role in this complex process, and C2C12 myoblasts have been reported to be a suitable model [27,28,29]. As shown in this study, miR-370-3p was sponged by lncMyoD. Furthermore, based on our observation, miR-370-3p promotes myoblast proliferation and inhibits its differentiation by repressing ACADSB.

Several studies have shown that muscle fiber-type composition is associated with fatty acids. Moreover, very long chain acyl-CoA dehydrogenase (VLCAD) and ACADSB dehydrogenase deficiency are common mitochondrial fatty acid β-oxidative metabolic disorder diseases [31,32,33]. The clinical manifestations of ACADSB- and VLCAD-deficiency can vary, such as developmental delay, muscular atrophy, hypotonia, and myasthenia gravis. Sara et al. reported that VLCAD deficiency caused oxidative fiber type I to be significantly upregulated in VLCAD^−^/^−^ mouse skeletal muscle [33]. These results imply that ACADSB may also play a crucial role in the fiber-type transition. Thus, we investigated fast fiber type- and slow fiber type-related genes after ACADSB was silenced through qRT-PCR analysis. As expected, the results showed that the expression of slow fiber-type markers was significantly upregulated, and the expression of fast fiber-type markers showed a downward trend. We also investigated the involvement of ACADSB in the fiber-type transition through immunofluorescence analysis. Consistent with the qRT-PCR results, the fast myosin-positive cell ratio was decreased after ACADSB inhibition compared with the NC group.

## 5. Conclusions

In the present study, we showed the important role of lncMyoD in C2C12 myoblast proliferation and differentiation. Overexpression of lncMyoD promoted myogenic differentiation and inhibited the formation of slow-twitch myofibers, whereas si-lncMyoD showed the opposite trend. Our results confirmed that lncMyoD directly targets miR-370-3p. Furthermore, our data showed that miR-370-3p sponges ACADSB directly, which also regulates myoblast differentiation. Therefore, we presented evidence that the lncMyoD-miR-370-3p-ACADSB axis is involved in C2C12 myoblast proliferation and differentiation and regulates fiber type transition. Our findings suggest that lncMyoD serves as a potential regulatory factor that influences myogenesis by competitively binding miR-370-3p and then inducing ACADSB.

## Figures and Tables

**Figure 1 genes-12-00589-f001:**
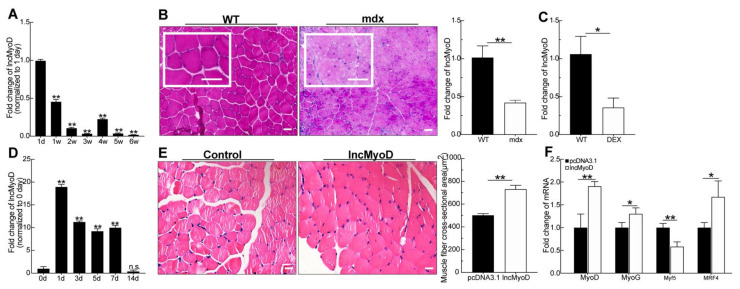
lncMyoD is a novel long-noncoding RNA (lncRNA) associated with myogenesis. (**A**) The expression of lncMyoD in postnatal mice at the indicated ages was detected by qRT-PCR. (**B**) H & E staining of mdx and WT group tibialis anterior (TA) muscle. qRT-PCR measured the expression levels of lncMyoD in C57B/L10ScSn-Dmdmdx/J (mdx) and wild type (WT) mice TA muscle. (**C**) TA muscles were isolated from 8-week-old C57B/L6 mice, which, after being treated with dexamethasone and RNAs, were extracted and used for qRT-PCR assay. (**D**) qRT-PCR measured the expression levels of lncMyoD during CTX-induced muscle regeneration. (**E**) Representative image from H&E staining of mice TA muscle after injection with pcDNA3.1 and lncMyoD vector for 1 month. (**F**) mRNA expression level of MyoD, MyoG, Myf5, and MRF4 in mice TA muscle injected with pcDNA3.1 and lncMyoD vector. Scale bars in (I), 100 μm and in (**B**,**E**), 20 μm in magnified (**B**). All PCR data were normalized to β-actin mRNA and represent means ± SD of three independent experiments. Data are means ± SD. *p*-values are obtained from independent-samples *t*-tests. * *p* < 0.05, ** *p* < 0.01.

**Figure 2 genes-12-00589-f002:**
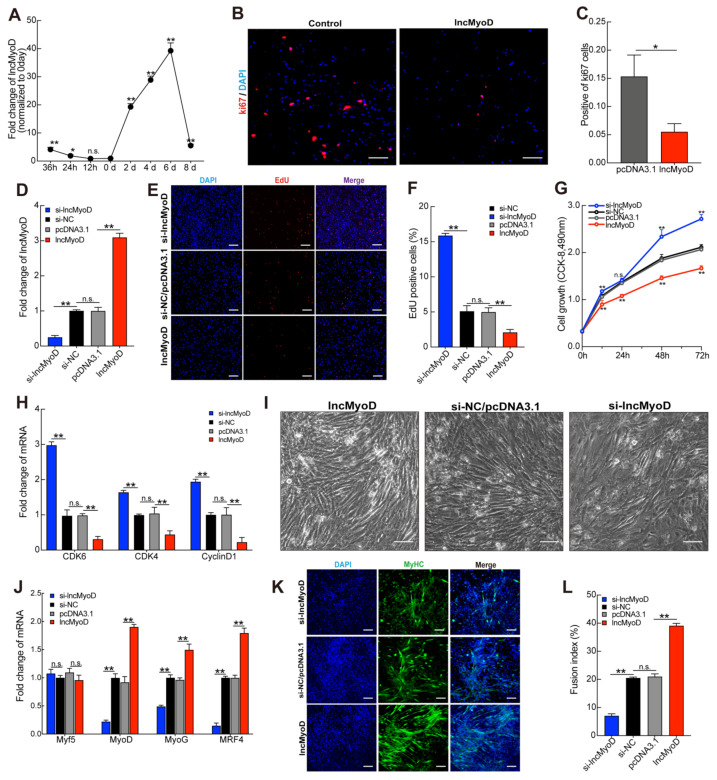
lncMyoD regulate myoblast proliferation and differentiation. (**A**) The expression levels of lncMyoD during C2C12 myoblast proliferation and differentiation. (**B**,**C**) The ki67 expression levels were obtained by immunofluorescence analysis after mice were injected with pcDNA3.1-lncmyod or pcDNA3.1. (**D**)After C2C12 myoblasts in GM (growth medium) were transfected with synthesized lncMyoD (pcDNA3.1-lncMyoD), pcDNA3.1, si-lncMyoD, and si-NC, the EdU proliferation assay (**E**,**F**) and CCK-8 assay (**G**) were used to determine cell proliferation. (**H**) The expression level of CDK6, CDK4, and cyclinD1 at 24 h of transfection was evaluated by qRT-PCR. (**I**) After transfection for 8 d, C2C12 myoblast differentiation was observed by bright-field microscopy. (**J**) qRT-PCR measured the expression levels of MyoD, MRF4, MyoG, and MyHC. (**K**) Immunofluorescence was used to analyze MyHC, (**L**) fusion index of myotube. Scale bars in (**I**), 100 μm and in (**B**,**E**,**K**) 50 μm. Data are means ± SD. *p*-values are obtained from independent-samples *t*-tests. * *p* < 0.05, ** *p* < 0.01.

**Figure 3 genes-12-00589-f003:**
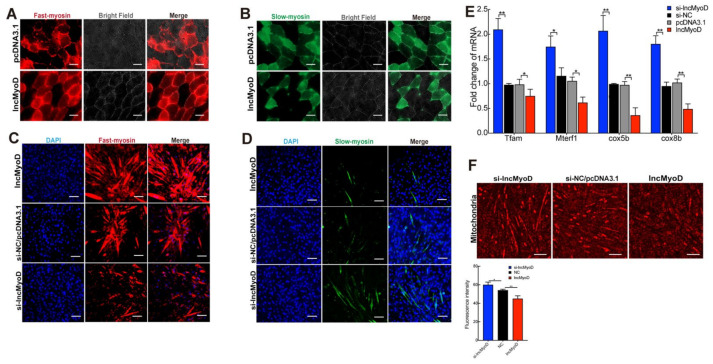
Increased expression of lncMyoD affects skeletal myofiber composition in vitro**.** pcDNA3.1-lncmyod and pcDNA3.1 were injected into mice via intraperitoneal injection of 4-wk-old mice for 28 d. (**A**,**B**) Immunofluorescent staining results demonstrated that lncMyoD significantly promoted fast-myosin fibers and suppress slow-myosin fibers within a cross-section of the gas, and qRT-PCR analysis the expression level of gene-related fast- and slow-twitch fibers. Differentiated C2C12 myotubes were transfected with lncMyoD (pcDNA3.1-lncMyoD), pcDNA3.1, si-lncMyoD, and si-NC for 8 d, and immunofluorescence was used to analyze myosin-fast–positive (**C**) myosin-slow–positive (**D**). (**E**) The expression levels of gene-related mitochondrial biogenesis and mitochondrial OXPHOS were measured by qRT-PCR. (**F**) Immunofluorescence staining using mouse anti-mitochondria antibody was performed in differentiated C2C12 myotubes. Scale bars, (**F**) 100 μm and (**A**,**B**,**D**,**E**) 50 μm. Data are means ± SD. *p*-values are obtained from independent-samples *t*-tests. * *p* < 0.05, ** *p* < 0.01.

**Figure 4 genes-12-00589-f004:**
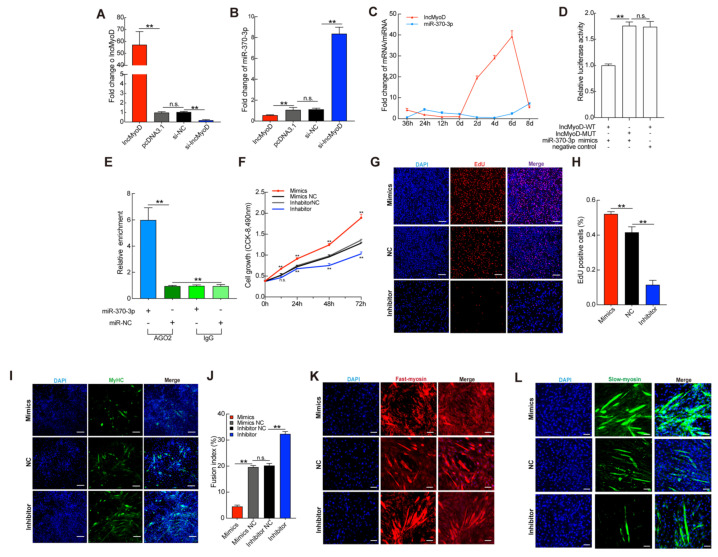
lncMyoD direct targeting miR-370-3p to mediate skeletal myofiber composition. When C2C12 myoblast differentiation was induced by differentiation medium (DM), the cells were transfected with miR-370-3p mimics, mimics NC, miR-370-3p inhibitor and inhibitor NC. (**A**,**B**) The expression of lncMyoD and miR-370-3p when C2C12 were transfected with lncMyoD, pcDNA3.1, si-lncMyoD, si-NC in DM at 8 days. (**C**) qRT-PCR measured the expression levels of lncMyoD and miR-370-3p during C2C12 proliferation and differentiation. (**D**) Luciferase assays revealed the suppressive effect of miR-370-3p on the activity of lncMyoD. (**E**) LncMyoD relative expression was determined by being pulled down with anti-AGO2. After C2C12 myoblasts in GM were transfected with synthesized miR-370-3p mimics, mimics NC, miR-370-3p inhibitor and inhibitor NC, CCK-8 assay (**F**) and EdU proliferation assay (**G**,**H**) at 24 h of transfection was evaluated. (**J**–**L**) After transfection for 8 d of C2C12 myoblast differentiation the immunofluorescence was used to analyze MyHC (**I**), fusion index of myotube (**J**). Differentiated C2C12 myotubes were transfected with miR-370-3p mimics, mimics NC, miR-370-3p inhibitor and inhibitor NC for 8 d, and immunofluorescence was used to analyze myosin-fast–positive (**K**) myosin-slow–positive (**L**). Scale bars in (**G**) 100 μm and in (**I**,**K**,**L**) 50 μm. Data are means ± SD. *p*-values are obtained from independent-samples *t*-tests. ** *p* < 0.01, n.s. no significant.

**Figure 5 genes-12-00589-f005:**
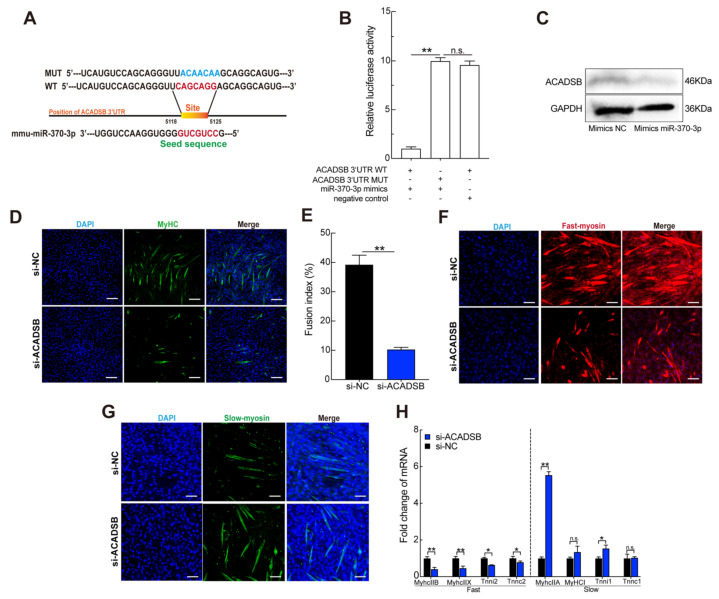
acyl-Coenzyme A dehydrogenase, short/branched chain (ACADSB) is a direct target of miR-370-3p during skeletal myogenesis. (**A**) Sequence alignment of miR-370-3p with 3′ UTR of LACTB. (**B**) Luciferase assays revealed the suppressive effect of miR-351-5p on the activity of ACADSB. (**C**) ACADSB protein expression level in miR-370-p mimics and control cells. (**D**,**E**) After transfection for 8 d, C2C12 myoblast differentiation immunofluorescence was used to analyze MyHC, fusion index of myotube, and immunofluorescence was also used to analyze myosin-fast–positive (**F**) and myosin-slow–positive (**G**). (**H**) The expression levels of gene-related fast-twitch fibers and slow-twitch fibers were measured by qRT-PCR. Scale bars in (**D**,**F**,**G**) 50 μm. Data are means ± SD. *p*-values are obtained from independent-samples *t*-tests. * *p* < 0.05, ** *p* < 0.01.

## Data Availability

This study did not report any data.

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
