# Peer review of "LncMyoD Promotes Skeletal Myogenesis and Regulates Skeletal Muscle Fiber-Type Composition by Sponging miR-370-3p"

_genes, 2021, doi:10.3390/genes12040589_

Round 1

Reviewer 1 Report

The study concerns LncMyoD and miR370-3p in in vitro growing myoblasts as well as in skeletal muscle, under various physiological and non-physiological conditions (such as induced injury). The role of LncMyoD and miR370 in skeletal muscle development and function has been previously studied and these studies need to be properly presented in the Introduction, not only extremely briefly mention. Next, the claim that the analysis of lncMyoD in skeletal muscle is novel has to be a bit calmed down, and again supported by already existing data.

Overall, the paper is sound, however, difficult to follow due to the overlap of the results and discussion. The best solution would be to separate these two parts. Show results and than discuss them. Nevertheless, the results presented are interesting and important for the understanding of muscle function. However, in order to be accepted manuscript needs to be improved and several issues clarified.

  1. Text needs to be carefully read. Such mistakes as in line 71 should be corrected, as well as the spaced added in appropriate positions, especially before or after the brackets.
  2. Introduction needs to be supplemented by the brief summary of the current knowledge on lncMyoD and miR370, as far as mesoderm development and skeletal muscles are concerned
  3. Material and methods need to include information about the permits to work with GMO
  4. What was the number in each mice group analyzed? The Authors mention 12 but only for lncMyoD overexpressing mice.
  5. Chapter 3.1 of Results is extremely dense and chaotic. In fact it is difficult to follow. Please describe each experimental system separately. mdx mice and mice subjected to skeletal muscle injury should be described next to each other - each od the models covers muscle regeneration. Common features should be emphasized. Muscle atrophy model should be described and discussed separately. The lncMyoD mouse model should be better described.
  6. Part of the text starting from line 206 contains very basic description of MRFs - it is not related to the results. Thus, it should be removed or better connected to the subject of the project. It is not even the discussion of the results.
  7. Figure 1 B. Quality of histological image of mdx muscle is poor. Should be replaced with the one in which centrally positioned nuclei would be clearly visible.
  8. Figure 1G and H. Images of the fibers are of very bad quality. I do not see neither MyoD nor MyoG localized within the nuclei. On the basis of this images I am not able to say that there is any difference in the expression/localization of these transcription factors. This should be better documented.
  9. Why primary myoblast have not been used? It should of interest to compare lncMyoD in myoblasts isolated from fast and slow muscles. Next, the C2C12 are not the perfect model for such studies. Primary cells should be used.
  10. Figure 3F should be replaced, it is hardly interpretable

Author Response

请参阅附件。

Reviewer 2 Report

It was a great pleasure reviewing the article entitled “LncMyoD promotes skeletal myogenesis and regulates skeletal 2 muscle fiber-type composition by sponging miR-370-3p” by Peiwen Zhang et al. The authors investigated the role for “Long noncoding RNA MyoD” in skeletal muscle development including proliferation and differentiation. Using a combination of invitro and in vivo systems, they have presented a good case that the lnc-537 MyoD-miR-370-3p-ACADSB axis is involved in proliferation and differentiation of skeletal muscle cells and regulates myofiber type transitions. In this regard, the experiments are done carefully, and the data is quite compelling. However, I have identified following issues with the manuscript that needs a serious attention from the reviewers.

  1. In figure 1G, the authors claimed that down-regulation of lncMyoD with siRNA oligos in single fibers decreased the satellite cell differentiation, which they tested indirectly through staining the MyoD and myogenin proteins. Staining on single fiber is unclear particularly, white arrows do not seem to indicate anything particular. I suggest the authors to perform immunostaining for any bonafied satellite marker such as PAX7 to directly show the enrichment of satellite cells in siRNA-lncMyoD condition as compared to controls. This will help reinforce that satellite cell differentiation is halted in lncMyoD down-regulated fibers.
  2. In figure 2K, the fluorescence brightness of the merged images is markedly different from the brightness of the individual DAPI and MyHC channels. The same is true for figure 4I. It would be nice to adjust the fluorescence contrast to make the figure panels look balanced.
  3. While describing the expression levels of lncMyoD in the Soleus (SOL) and Tibialis Anterior (TA) muscles, the authors stated that lncMyoD is highly expressed in SOL that TA (page 9: line 322). This actually is not a true statement as the data presented in supplementary figure 1C and Figure 3 show high expression in the TA muscles. The authors are suggested to remove this confusion.
  4. Figure 3F, mitochondria staining does not seem to support qPCR data presented in Figure 3E showing increased expression of oxidative genes. Immunofluorescence (red) signal look much lower visually in silncMyoD condition than pcDNA control or lncMyoD conditions. It would be great to provide quantification on the stained images presented in Panel F.
  5. On page 11, line 408, the supplementary figure 3A is not correctly referred in the text, instead of Supplementary figure 3A, the authors have referred to main figure 3A, which is obviously inappropriate. There are more issues of misreferred data in the text, for example, on page 11, line 438, it is actually Supplementary Figure 2E not 2D, and page 11, line 446, it is Supplementary Figure 2F, not 3E. The authors are asked to revisit the text extensivity to eradicate the mislocation of the data. 
  6. Figure 4C, x-axis scale is confusing, it should go from hours to days….and so on….
  7. Result section on page 12, line 469 should be numbered as 4.5 not 4.7.

Round 2

Reviewer 1 Report

The Authors introduced required corrections and I have to agree that in the current form the work is much more clear and accessible. However, even if I am not a native speaker I still see many gramma errors to be corrected and sentences to be clarified. For example, line 57-58 - sentence needs to be clarified; line 60 bind should be replaced by binds (in fact this is a common mistake made by the Authors, manuscript should be checked for singular plural forms of the verbs); lines 62, 71, 72 - sentences to be corrected; 186 - "we performed three mouse models" (!). And many many more mistakes. English editing is absolutely necessary. Next, Figure 1B needs to be corrected as required. In the current version the inserts do not present higher magnifications. Please include phots which are truly magnified with magnification bars inside.
